# Event-driven architecture and intelligent decision tree facilitated sustainable trade activity monitoring model design

Yixian Wen[1]*, Suo Zhang[2], Sisi Zhang[1]

1 School of Business, Hunan Institute of Technology, Hengyang, Hunan, China, 2 School of Intelligent Manufacturing and Mechanical Engineering, Hunan Institute of Technology, Hengyang, Hunan, China

* wenyixian@hnit.edu.cn

## Abstract

This paper introduces a groundbreaking monitoring model tailored for sustainable trade activity surveillance, which synergistically integrates event-driven architecture with an intelligent decision tree. Confronting the constraints of conventional trade monitoring approaches that falter in adapting to the intricate and ever-changing market landscape, our model innovatively establishes an efficient, adaptable, and sustainable monitoring framework. By embedding an intelligent decision tree, it enables dynamic resource allocation, thereby optimizing operational efficacy. Initially, we devise rules that align data injection and processing velocities, ensuring expedient data processing. Subsequently, we implement an optimal binary tree decision-making algorithm, grounded in dynamic programming, to achieve precise allocation of elastic resources within data streams, significantly bolstering resource utilization. Throughout the monitoring continuum, the model employs intelligent agents to assess resource status in real-time and dynamically adjusts resource allocation strategies triggered by events, prioritizing the seamless execution of pivotal trade activities. Empirical findings underscore the model's superiority across critical metrics, including data accumulation efficiency, processing latency, resource utilization, and throughput. Specifically, it attains an average data accumulation value of 15.46, curtails latency by 14.67%, achieves an average resource utilization of 60.29%, and registers a throughput of 336.5 Mbps. Consequently, the model not only exhibits rapid responsiveness to market fluctuations and curtails resource energy consumption but also fosters a harmonious equilibrium between economic gains and environmental preservation, ensuring the uninterrupted operation of trade activities.

## 1. Introduction

Under the wave of globalization and informatization, trade activities are evolving at an unprecedented speed and complexity. The interconnected nature of international trade is increasingly pronounced, with the rise of cross-border e-commerce

**Data availability statement:** All files are available from the database(Anneal: https://zenodo.org/records/14176982, doi: 10.5281/zenodo.14176982 UN Comtrade: https://www.greenpolicyplatform.org/platforms/un-comtrade-database, doi: 10.4236/ojbm.2021.96147 WDI: https://www.kaggle.com/datasets/theworldbank/world-development-indicators, doi: 10.4236/ojpp.2014.42015 IMF: https://zenodo.org/records/7990528, doi: 10.5281/zenodo.7990528 OECD: https://zenodo.org/records/5762158, doi: 10.5281/zenodo.5762158).

**Funding:** This work is funded by 2024 Hunan Natural Science Foundation Project "Data Elements Embedded in Entity Value Chain: Empowerment and Risk", the project number is 2024JJ6199.

**Competing interests:** The authors have declared that no competing interests exist.

facilitating the global movement of goods and services. However, this high degree of interconnectedness and the rapidly changing market environment present significant challenges to the sustainability of trade activities.

The sustainability of trade activities has become a crucial metric for assessing the economic health and social responsibility of a country or company [1]. This concept extends beyond environmental protection to encompass social justice, economic viability, and legal compliance. Specifically, sustainable trade requires companies to not only rigorously control carbon emissions—through measures such as employing clean energy and optimizing logistics routes to minimize the carbon footprint of transportation—but also to ensure the protection of workers' rights and interests. This includes providing a safe working environment, fair remuneration, and opportunities for career development. Additionally, enterprises engaged in international trade must adhere to the international legal framework, which includes principles of fair trade and intellectual property protection. They should also actively seek ways to reduce energy consumption and resource waste to achieve a win-win outcome in terms of economic benefits and environmental protection. However, traditional methods of monitoring trade activities often rely on fixed processes and rules, which struggle to adapt to the rapidly evolving market environment and diverse business needs. Traditional monitoring approaches, which depend on manual operations and static rule bases, are inadequate for managing large volumes of data and complex scenarios [2]. As technologies such as big data and artificial intelligence continue to advance, there is a growing need for more efficient and intelligent monitoring solutions. Such systems must be capable of real-time responsiveness, flexible adjustment of monitoring strategies, and provision of scientifically accurate decision support for decision-makers.

Although some studies have attempted to apply emerging technologies to trade activity monitoring, existing approaches remain inadequate in coping with the complexity and dynamism of modern market environments. In particular, traditional monitoring models struggle with stream processing applications involving multi-source inputs and multi-directional outputs due to rigid architectural designs and limited scalability. This often leads to mismatches between data ingestion and processing rates, resulting in increased processing delays and inefficient resource utilization. Such limitations not only reduce the effectiveness of trade monitoring but also hinder the achievement of dual goals: economic efficiency and environmental sustainability. Therefore, there is an urgent need for a new monitoring model capable of real-time market sensing, adaptive resource allocation, and intelligent decision-making to overcome the constraints of traditional methods and enhance the overall sustainability of trade activities.

In this context, the advent of Event-Driven Architecture (EDA) [3] and intelligent decision tree technology [4] offers novel solutions for monitoring sustainable trade activities. EDA, characterized by its asynchronous and loosely-coupled nature, allows systems to respond to various events in real time and promptly deliver relevant information. This architectural model enables systems to adapt flexibly to different

situations and achieve efficient monitoring [5]. Meanwhile, intelligent decision trees represent decision-making processes through a tree structure, analyzing and predicting outcomes using historical and real-time data, thereby providing a scientific basis for decision-making.

In the realm of stream processing applications with multiple inputs and outputs, traditional development models using a single distributed data stream engine exhibit significant limitations. These limitations manifest not only in the rigidity of the architectural design, which hampers flexibility in adapting to evolving business requirements, but also in the lack of scalability, which impedes the effective expansion of processing capacity as data volumes increase. When applying distributed stream processing technology based on event-driven architecture to trade activity monitoring scenarios, a notable mismatch can occur between the data injection rate and the data processing rate. This discrepancy is particularly pronounced when the number of data sources fluctuates or when data distributions change significantly, leading to increased processing delays and inefficient resource utilization.

To address this issue, this paper examines the root causes of the inconsistency between data injection and data processing rates. It integrates the advantages of dynamic planning offered by the optimal decision tree algorithm with the event-driven architecture to design a sustainable trade activity monitoring model. This model is capable of real-time market sensing, flexible resource allocation adjustment, and intelligent decision-making. Not only does this model swiftly respond to diverse business demands, but it also ensures the stable operation of key trade activities by optimizing resource allocation strategies under limited resources. Consequently, it significantly enhances the overall performance and resource utilization efficiency of the monitoring system while reducing energy consumption.

The specific contributions of this paper are as follows:

Achieving Data Consistency in Event-Driven Distributed Stream Processing: This paper establishes rules to align upstream and downstream data injection speeds with data processing speeds. This alignment enables the resource manager to detect state changes in upstream and downstream applications, allowing for timely adjustments in the resource requirements of stream processing applications.

Development of the Optimal Binary Decision Tree Algorithm Based on Dynamic Programming (dp-OBDT): The algorithm computes the optimal classification tree through exhaustive search. It utilizes root node allocation as the optimal allocation to store or update the corresponding entries, with the lower bound determined by the misclassification score and considering the depth and feature node limitations of the computed subtree. Overlapping between trees is leveraged to reduce the computation time of the optimal decision tree to manageable levels.

Implementation of Optimal Allocation of Elastic Data Flow Resources Using the Optimal Binary Decision Tree Algorithm: Based on the optimal binary decision tree algorithm, the model optimally allocates elastic data flow resources. Actions are initiated and recorded in the command matrix based on the decision tree's depth and the number of feature nodes in the cache. The resource allocation module of the resource manager then reserves computing resources and makes new decisions based on the updated information from the command matrix and the intelligent agent, thus facilitating sustainable trade activity monitoring.

This paper will introduce the current research status of time-driven architectures and their application prospects across various fields in Section 2, along with related research on decision trees. Section 3 will detail the streaming data processing framework developed in this paper and the event-driven distributed streaming processing model based on the optimal binary decision tree. Section 4 will present experimental results, discussing the performance of the optimal binary decision tree and event-driven architecture, comparing them with existing decision trees, and analyzing their impact on elastic resource allocation in data streams. Section 5 will conclude with a summary of the performance of the proposed schemes and outline future research directions.

## 2. Related works

### 2.1 Event driven architecture

In 1996, researchers introduced a theory of design architecture grounded in service-oriented concepts [6]. Service-Oriented Architecture (SOA) is a coarse-grained, loosely coupled framework where services communicate through well-defined interfaces, abstracting away underlying programming interfaces and communication models. This approach allows architects to conceptualize applications in a modular fashion, breaking them down into loosely coupled, coarse-grained components that can be distributed, deployed, and combined as needed. For medium and large brokerage firms, adopting SOA principles can facilitate the segmentation of core business systems into distinct components, such as trading systems, core systems, accounting systems, and position management systems. However, challenges arise when a user initiates a securities subscription transaction, necessitating notifications and responses from all these systems simultaneously. The trading system must interact with numerous systems at once to manage these transactions effectively. To address these challenges, the industry has proposed integrating Event-Driven Architecture (EDA) with SOA [6].

EDA enhances SOA architectures by introducing an event-based mechanism, enabling systems to sense and respond to business events rapidly [7]. EDA employs an asynchronous publish-subscribe model, allowing for flexible one-to-many distribution. In this model, an event is published to a central broker, and multiple subscribing systems can detect and react to the event. EDA builds upon SOA principles by adding an asynchronous communication pattern that enhances SOA's ability to handle complex, heterogeneous systems. The integration of SOA and EDA focuses on improving event processing capabilities within SOA and facilitating asynchronous communication across diverse systems.

Currently, EDA is widely adopted by medium and large Internet companies both domestically and internationally. For these companies, rapid business development often involves core business systems utilizing multiple technology stacks. EDA is particularly suited for facilitating communication and interaction among complex, heterogeneous systems [8]. Additionally, many non-core services do not require synchronous request-response mechanisms. By leveraging EDA combined with distributed messaging middleware, asynchronous interactions can be effectively implemented, enhancing functionalities such as decoupling, peak shaving, and maximizing the asynchrony of messaging middleware.

Shi et al. [9] suggests using EDA-based stream processing to address challenges associated with multiple data sources and processing outputs. However, significant fluctuations in data sources can lead to processing delays and reduced system resource utilization. Issues such as low connectivity or intermittent disconnections of devices like sensors can disrupt data flow, while dynamic changes in data distribution characteristics can cause biases and delays in subsequent processing. To overcome these challenges, distributed stream processing applications based on EDA must be capable of adaptively allocating computational resources, ensuring consistency between data injection and processing, and achieving efficient resource utilization.

### 2.2 Decision trees

The decision tree algorithm is one of the most widely used methods in data mining. It focuses on categorization by identifying similarities and differences among data points. Each internal node of the tree represents a test on an attribute, with the test results corresponding to the branches, while the leaf nodes are labeled with classes, and the root node is at the top of the hierarchy. Decision trees leverage historical and real-time data for analysis and prediction, enabling efficient computational resource allocation [10]. Schidler and Szeide [11] proposes a method for constructing decision trees with a maximum depth of $k$, aiming to minimize misclassification scores. This approach uses $k$ as a constraint and models the problem within a Mixed Integer Programming (MIP) framework with a fixed number of variables. An MIP solver is then employed to find the optimal decision tree. This classification method simplifies training the optimal decision tree with hyperplane classification, making it as straightforward as training a corresponding univariate decision tree, thus representing a significant advancement over heuristic methods for multivariate decision tree problems. Liu et al. [12] introduces the

BinOCT algorithm, which optimizes a depth-constrained approach specifically for handling numerical data. To manage numerical data, the decision tree must recognize thresholds used to segment the data points effectively. Zhang et al. [13] presents the MI model, demonstrating through experimental results that this method significantly enhances the processing of numerical data and performs well on both small and large datasets. From a modeling perspective, an advantage of MIP-based algorithm optimization is the ease with which linear constraints or additional optimization criteria can be incorporated. Tang et al. [14] utilizes this advantage to formalize a learning problem while considering prediction fairness.

Currently, decision tree research primarily focuses on heuristic algorithms, with classic and widely used examples including ID3 [15], C4.5 [16], and CART [17]. Heuristic methods are capable of producing results quickly; however, they do not guarantee global optimality and often suffer from poor interpretability. As both algorithmic technology and hardware conditions have advanced, researchers have increasingly explored optimal decision tree algorithms. These algorithms generally involve integer programming, constraint programming, and techniques specifically tailored for decision tree optimization. Zhang et al. [18] introduces a mixed integer programming approach that performs well on smaller datasets. Demirović E and Stuckey [19] develops a mixed integer programming method to obtain optimal decision trees by presetting the tree depth, creating variables to represent node predicates, and incorporating constraints for tree computation. Roy and Chakraborty [20] presents a novel mixed integer programming approach based on Support Vector Machines (SVM) and multiple plane techniques to optimize multivariate decision trees. Aglin et al. [21] introduces a caching technique to save trees generated during subtree computation for reuse in later calculations and designs the DL8 algorithm, which lays a foundation for subsequent optimal decision tree algorithms. Building on this work, McTavish et al. [22] proposes the DL8.5 algorithm, which incorporates numerous optimization results, making it one of the most effective optimal decision tree algorithms available today.

Despite these advancements, optimal decision tree algorithms must consider all possible decision trees to ensure optimality, leading to substantial computational demands. Over the years, the development of optimal decision tree algorithms has incorporated general optimization techniques such as mixed integer programming and constraint programming, as well as caching and searching techniques tailored to decision tree characteristics. These improvements have significantly reduced computation time. Nevertheless, compared to heuristic algorithms, current optimal decision tree algorithms still exhibit lengthy run times. Consequently, further optimization techniques are needed to reduce the running time of optimal decision tree algorithms in applications such as detecting sustainable trade activities.

## 3. Methods

Event-driven distributed stream processing applications are crucial for trade activity stream processing platforms. They represent not only a technical architectural innovation but also a significant advancement in enhancing trade efficiency and intelligence. These applications integrate seamlessly into the global trade ecosystem, capturing real-time data flows from various channels, including order generation, cargo transportation, and payment confirmation. Each event in the system carries critical time-stamped information, which ensures the timeliness and accuracy of data processing.

In the event-driven model depicted in Fig 1, the streaming data, illustrated by solid lines with arrows, conveys dynamic information regarding the life cycle of trade activities. This data is persistently stored and efficiently retrieved by the distributed logging system, represented by the cylindrical shape. The components in the application layer, such as order processing, logistics tracking, and financial analysis—depicted by the final rounded rectangle—collaborate closely to analyze the raw streaming data in real-time. They update the status and trigger corresponding trade decisions or operations based on the insights gained. $A_{11}$ and $A_{12}$ are the applications that represent the processing of raw stream data, which we will later denote as the application that processes the resource manager data stream in Fig 2. $A_3$ As another distributed stream processing engine, by obtaining the output data of $A_2$ and then analyzing and processing it, its processing results can be directly output to $A_3$ and $A_4$.

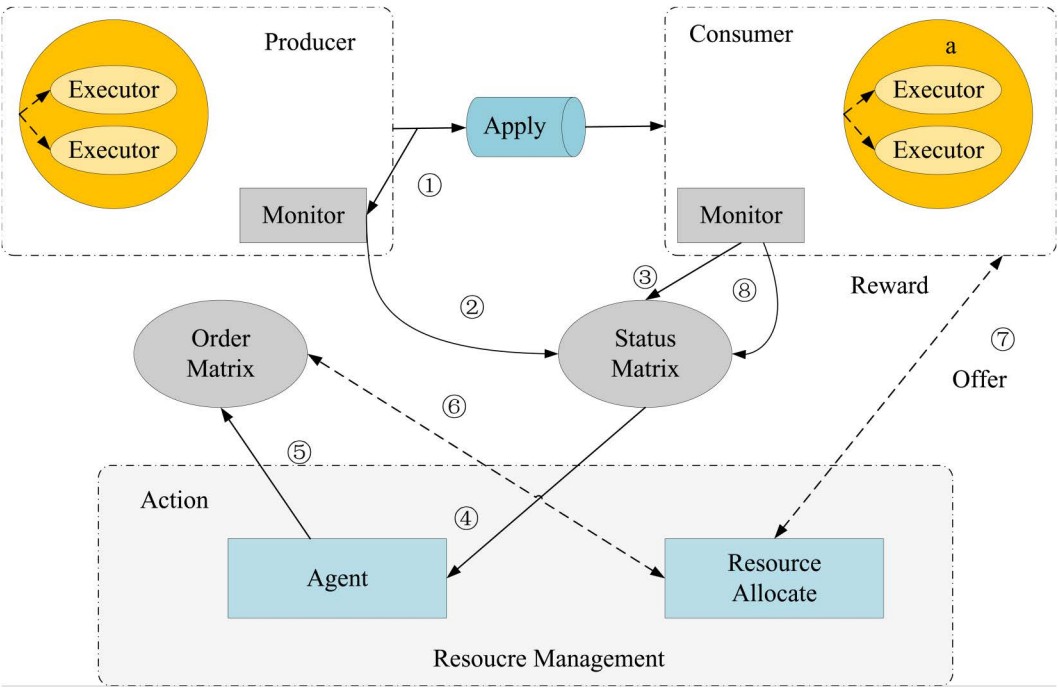

**Fig 1. Event-driven data flow processing.**

**Fig 2. Dynamic resource allocation process.**

At the resource allocation level, the right-angle rectangle represents the resource manager, which collects real-time state information from each application through interactions with the application layer. Utilizing advanced algorithms, the resource manager analyzes current resource usage and predicts future demand, dynamically adjusting the allocation of computing resources. This ensures that processing capacity remains efficient and stable even during peak trade periods, thereby guaranteeing the continuity and responsiveness of trade activities. This mechanism not only optimizes resource utilization but also significantly enhances the overall efficiency and competitiveness of trade operations.

### 3.1 Data consistency processing

First for a query in a trade activity, the consistency of its input and output data needs to be processed. Suppose that $W$ is a collection of micro-batches of processing intervals $W_i = \{(W_{i,j})\}$. Each processing interval $W_i$ is associated with an application $a_i$, which consists of a sequence of multiple iterations $W_{i,j}$, each iteration $W_{i,j}$ is defined as a time interval $w$, and the application $a_i$ obtains the measurements collected during this interval, which are collected at a predefined time cutoff, and

for each application $a_i$ collects, at each timestamp $m_{i,d}$, the number of tuples that were generated or processed during the interval $[m_{i,d-1}, m_{i,d}]$, where $d \in \{1, 2, ..., n\}$.

Upstream applications generate data, downstream applications consume data, and there is a flow between upstream and downstream applications. Let $D_i$ be the set of data produced by the application $a_i$, which is unbounded, and $D_{i,d}$ represent the set of data produced by the application $a_i$ in the time interval w. For Fig 2 contains a stream $e_{ij}$ between a producer application $a_i$ and a consumer application $a_j$. Data $a_i$ is generated at the detection point $m_0, m_1, m_2$ and this data is accumulated to obtain the data generated by $a_i$ during the processing interval $w_0$ as $D_{i0}$.

At the moment $t_0$, the processing of the data $a_j$ starts, the size of the injected data at $a_j$ is set to $l_{j1}$ and when = $l_{j1}D_{i0}$ is satisfied. The system allocates the necessary resource quota for $a_j$ based on the injected data to ensure that the processing of $a_j$ is not delayed.

In a processing interval $w$, the upstream application of $a_j$ generates consumes data. Assuming that each instance $a_i$ of $a_{i,k}$ generates $D_{ik}$ data, the speed rate is denoted as:

$$speed = D_{ik}/w \tag{1}$$

Then the rate at which $a_i$ generates data is written as:

$$speed_i = \sum_{k=1}^{n} D_{ik}/w \tag{2}$$

The downstream application $a_j$ needs to consume the data generated by the producer during the next processing interval and its data processing speed can be calculated as shown in Equation (3):

$$speed_j = \sum speed_i. \tag{3}$$

The data injection rate should be consistent with the data processing rate. For applications $a_i$ and $a_j$, if $speed_i = speed_j$, then the data injection speed is consistent with the data processing speed.

## 3.2 Event-driven distributed stream processing based on optimal binary trees

In the process of resilient resource allocation for event-driven stream processing-based applications, it is essential to adjust the number of parallel instances of an application in response to changes in the data injection rate, as the number of instances directly correlates with resource usage. The resource manager monitors the data generation speed of the upstream application and sends this information, along with the current state of the downstream application, to the decision-making body, i.e., the intelligent agent. Upon receiving the data injection rate and the application state, the intelligent agent decides whether to scale the resources up or down, aiming to optimize the application's service level. This optimization ensures that the application's latency closely matches the processing interval of the data stream. If the application's processing time exceeds one processing interval, it indicates insufficient resources, leading to data accumulation and increased latency. Conversely, if the processing time is significantly shorter than the processing interval, resource usage is excessive, despite meeting the delay requirements. The resultant execution information of the application is fed back to the intelligent agent to inform subsequent decisions.

This section, therefore, introduces the dp-OBDT framework, designed to identify the optimal classification tree structure through an intelligent exhaustive search strategy. It models the decision-making process as a tree structure and leverages both historical and real-time data for analysis and prediction. During the search, the algorithm carefully designs traversal strategies to exploit the overlapping substructure of the tree and avoid unnecessary computation of suboptimal trees. This effectively keeps the time complexity of finding the optimal decision tree within a reasonable range. Specifically, the

algorithm caches the boundary conditions of subtrees and the characteristics of optimal root nodes under constraints of feature depth and node count, thereby reducing redundant computations and improving search efficiency.

We abstract the processing mechanism of streaming data as a mapping mechanism from subtrees to a set of cache entries. Each cache entry details the subtree boundary conditions and optimal root node characteristics under constraints of depth and number of feature nodes. These details include the root node characteristics, the counts of the left and right child nodes' feature nodes, and the misclassification cost. Initially, the cache is empty but becomes enriched as the algorithm progresses. Given that this method focuses on analyzing tree structures with a depth of no more than three, we strategically avoid caching the bottom decision tree, significantly reducing the cache burden and optimizing space utilization. Additionally, based on the technical details described later, the retrieval efficiency of subtrees in the cache is extremely high. After a subtree has been exhaustively searched, its optimal information is solidified in the cache. If a decision tree meeting the requirements is found within the given boundaries, the corresponding subtree is marked as optimal and updated or added to the cache entry with the root node assignment scheme as the optimal configuration. A lower bound based on the misclassification score is recorded to ensure that this record strictly adheres to the preset depth and feature node constraints. Furthermore, when the algorithm identifies that minimizing misclassification cost can be attained by decreasing the number of nodes (while adhering to or surpassing the maximum gain potential within node constraints), we implement a mechanism to dynamically generate extra cache entries. This improvement bolsters the cache's flexibility and adaptability.

Let $T(D, d, n)$ be the misclassification score of the optimal decision tree for the dataset D with depth restriction d and number of nodes restriction n. If there exists n'<n satisfying = $T(D, d, n)T(D, d, n')$, then Equation (4) can be obtained:

$$T(D, d, i) = T(D, d, n),\ i \in [n', n]$$

(4)

If the depth used does not exceed the maximum allowed depth, a similar approach is used to populate the entries. It should be noted that a given branch or dataset may only be thoroughly explored once during the algorithm. Depending on the subtree representation used in the cache, the next time the branch or dataset is encountered, the corresponding solution will be efficiently retrieved from the cache.

When no decision tree is found within the upper bound, the lower bound on the number of sub-tree misclassifications is at least 1 larger than the upper bound. At this point, it can be shown that there exists a stronger lower bound, $LB(D, f, d)$ is the optimal lower bound that can be found in the cache which obviously satisfies Equation (4). Let $T(D, d, n)$ be the lowest misclassification score for a given dataset D under the constraints of maximum depth of d and maximum number of nodes of n. From this a stronger low bound can be introduced:

$$LB(D, f, d) = LB(D(f', d-1, n_{left})) + LB(D(f, d-1, n_{right}))$$

(5)

$$RLB(D, d, n) = \left\{ \begin{array}{l} \min\{LB(D, f, d, n_{left}, n_{right})\} \\ f \in F \cap n_{left} + n_{right} = n-1 \end{array} \right.$$

(6)

According to Equation (5), the lower bound $RLB$ calculates all the assigned combinations of feature nodes and selects the smallest among them as the value of $RLB$. Therefore, the misclassification score of any decision tree cannot be lower than $RLB$. Combining $RLB$ with the upper bound $UB$ gives:

$$T(D, d, n) \geq \max\{RLB(D, d, n), UB + 1\}$$

(7)

Once the lower bound is computed, it is recorded in the cache along with the constraints on the depth of the decision tree and the number of feature nodes. As shown in Fig 2, the process is divided into eight main steps. For the cache content, the application generates streaming data, and its monitor regularly reports information about the streaming data

to the state matrix, where it is stored. After a decision is made, an action is initiated and written to the command matrix. The resource allocation module of the resource manager then reserves computational resources based on the latest action information from the command matrix and sends a command to the stream data processing engine to adjust the resources accordingly. The consumer uses the updated amount of resources to process the current micro-batch of data. The delay of this processing is then fed back to the state matrix, enabling the intelligent agent to make a new decision.

In terms of computational complexity, the primary overhead of the dp-OBDT algorithm arises from the exhaustive search over all possible decision tree structures. Although techniques such as caching and exploiting structural overlap help reduce the computational burden, the time complexity can still grow exponentially when applied to large-scale datasets. Nevertheless, compared to traditional optimal decision tree algorithms, dp-OBDT significantly reduces computation time through carefully designed traversal strategies and an efficient caching mechanism, enabling the discovery of near-optimal decision tree structures within a reasonable time frame.

## 4. Experiments and analysis

In this section, we analyze the performance of the proposed model and validate its efficacy by comparing it with the heuristic algorithm CART [17], the DL8.5 algorithm [22], and the Random Forest algorithm [23]. Additionally, we assess the model's impact on resource allocation performance through a series of ablation experiments. To ensure the reliability and stability of the experimental results, each experiment was repeated 1,000 times, and the average was taken as the final result.

### 4.1 Datasets

We select five datasets for experimental testing and set multiple parameter instances under different datasets, as shown in Table 1.

Table 1 summarizes the datasets used in our analysis. The Anneal dataset (http://archive.ics.uci.edu/ml/index.php) is commonly employed for classification or regression tasks in machine learning and data mining. The UN Comtrade dataset (https://comtrade.un.org/) contains comprehensive merchandise trade statistics from official sources, covering over 97% of global trade in goods. The WDI dataset (https://download.csdn.net/download/T0620514/88571060) includes data on various indicators such as GDP, GDP per capita, trade volume, investment, demographics, education, and health across multiple domains. The IMF dataset (https://data.imf.org/en) provides a range of economic data, including balance of payments, foreign exchange reserves, exchange rates, and interest rates. The OECD dataset (https://data.oecd.org/) offers information on import/export volumes, trading partners, commodity classifications, and other related areas. Based on these datasets, we present the parameter settings, where |D| represents the number of instances, |F| denotes the number of binary features, and |C| indicates the number of categories in the dataset.

### 4.2 Evaluation index

For the binary decision tree there are four kinds of classification results, which are true example TP, true inverse example TN, false positive example FP and false inverse example FN. the specific definitions are shown in Table 2:

**Table 1. Parameter settings.**

| data set | |D| | |F| | |C| |
|---|---|---|---|
| Anneal | 812 | 93 | 2 |
| UN Comtrade | 1055 | 81 | 2 |
| WDI | 688 | 89 | 2 |
| IMF | 296 | 95 | 2 |
| OECD | 1484 | 89 | 2 |

**Table 2. Confusion matrix.**

| the real situation | Example of a forecast | Predicting counterexamples |
|---|---|---|
| standard practice | TP | FN |
| counter-example | FP | TN |

Equation (8) [24] calculates the precision:

$$precision = \frac{TP}{TP + FP} \tag{8}$$

To evaluate the performance of the event-driven architecture in the trade activity monitoring model, this paper employs several metrics——data accumulation, resource utilization, throughput. Data accumulation reflects system delays by measuring the queue length of unprocessed requests at a given time—larger backlogs indicate slower processing and longer response times. Resource utilization captures how efficiently system resources (e.g., CPU, memory) are used over time, with higher usage indicating better resource efficiency. Throughput evaluates the system's capacity to handle concurrent requests and is estimated by dividing the number of simultaneously processed requests by the average response time, yielding the number of requests handled per second.

## 4.3 Runtime comparison

Since the DL8.5 algorithm is currently recognized as the leading decision tree algorithm in terms of running time, this section conducts a performance comparison between the dp-OBDT algorithm proposed in this paper and the DL8.5 algorithm. Specifically, we compare the running times of the dp-OBDT algorithm and the DL8.5 algorithm at tree depths of 4 and 5. This comparison aims to demonstrate the improvements made by the dp-OBDT algorithm in addressing the data consistency problem.

According to the runtime results presented in Fig 3, the method proposed in this paper demonstrates a significant performance improvement over the DL8.5 algorithm. This is noteworthy given that DL8.5 itself shows considerable performance gains compared to other optimal classification tree techniques based on integer and constraint planning. The experimental results highlight the advantages of designing specialized decision tree optimization algorithms as opposed to relying on general-purpose methods. While both DL8.5 and the method in this paper leverage the structural properties of decision trees, the proposed method incorporates additional techniques to further exploit these properties. The observed reduction in runtime enhances the practical application of the optimal decision tree methodology, particularly in tuning scenarios.

## 4.4 Performance of dynamic elastic allocation

To analyze the performance advantages of the dp-OBDT algorithm for resource elastic allocation, this section conducts comparative experiments with two decision tree algorithms: the CART algorithm and the random forest algorithm. To ensure comprehensive experimentation, this section incorporates hyperparameter tuning for the optimal binary decision tree based on dynamic programming to examine the effects of various tuning options. The first approach involves heuristically derived trees, where a single parameter is varied while keeping the parameter values fixed to those generated by the CART decision tree. The second approach involves heuristically obtained decision trees that establish upper bounds for tuning. Specifically, for a tree constructed by CART with depth $d$ and $n$ feature nodes, tuning is performed for depths $d \in [1, d]$ as well as $n \in [d, 2^d - 1]$. The third one is tuned from the depth $d \in [1, 4]$ and the number of feature nodes $n \in [d, 2^d - 1]$.

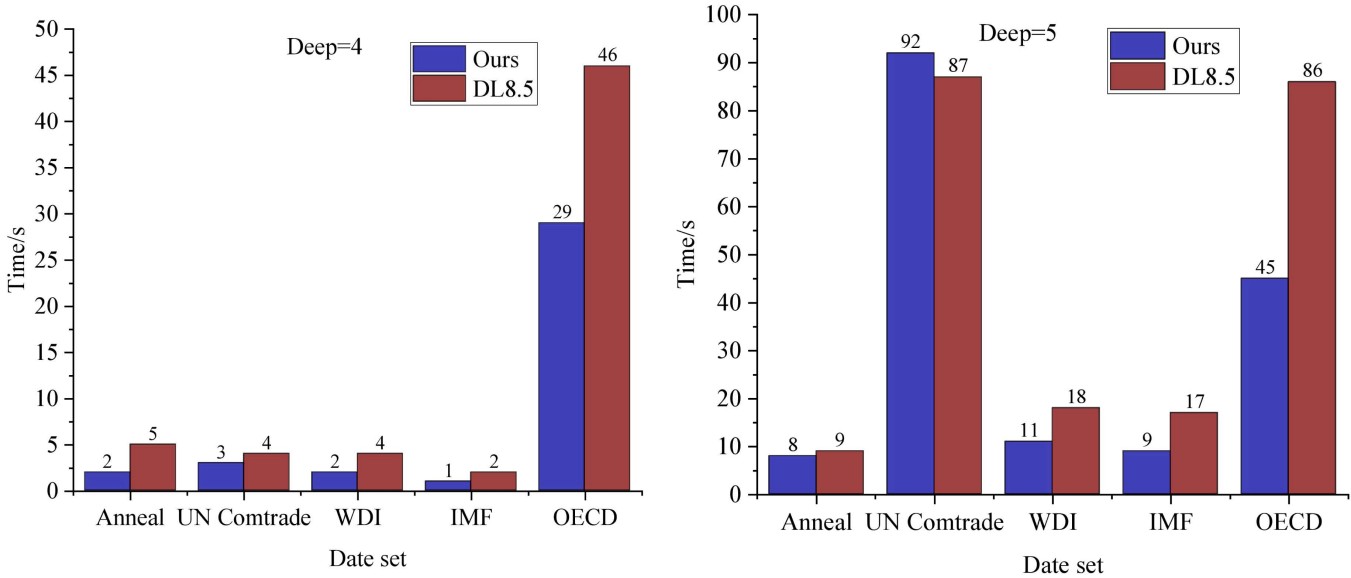

**Fig 3. Comparison of running time under different parameter settings.**

Based on the comparison of runtime performance presented in Fig 3, it is evident that runtime varies significantly with tree depth for both the UN Comtradel dataset and the OECD dataset. To further analyze the performance across these two datasets, the tuned CART algorithm is compared to the results obtained with the dp-OBDT algorithm, as illustrated in Fig 4. The vertical axis in Fig 4 represents the difference in prediction accuracy and runtime between the dp-OBDT algorithm and the CART algorithm, while the horizontal axis indicates tree depth.

Sub-Figure (1) of Fig 4 displays the results for the UN Comtradel dataset. Despite the higher training accuracy of the optimal decision tree, the performance of the dp-OBDT algorithm is roughly comparable to that of the CART algorithm. The observed discrepancy in test set performance suggests that the CART-generated tree structure may not be optimal for this dataset. When parameter choices are more flexible, performance differences become more apparent. As shown in subplot (2) of Fig 4, the parameter selection strategy applied to the OECD dataset yields superior results for most datasets, with the dp-OBDT algorithm achieving higher accuracy on the test dataset when all parameter options are considered.

Although the dp-OBDT algorithm generally requires more time to execute compared to the CART algorithm, this difference is acceptable given that decision tree algorithms are typically used offline. Overall, while the dp-OBDT algorithm takes longer to compute, this additional time is justified by improved model generalization. For offline algorithms where generalization is crucial, the enhanced performance provided by the dp-OBDT algorithm represents a valuable improvement.

To further evaluate the balanced performance of the proposed algorithm in terms of prediction accuracy and runtime on the UN Comtradel dataset, this section employs the random forest algorithm from scikit-learn, a machine learning package in Python, for tuning and comparison. Random forests, which consist of multiple decision trees, typically provide more accurate predictions than individual decision trees. However, these models are generally more complex and less interpretable.

In the experiments conducted, the size of the forest was adjusted, and the results are illustrated in Fig 5. The random forest algorithm demonstrates higher prediction accuracy on the training dataset. Nevertheless, the performance gap between the optimal decision tree algorithm and the random forest algorithm is minimal across nearly half of the datasets.

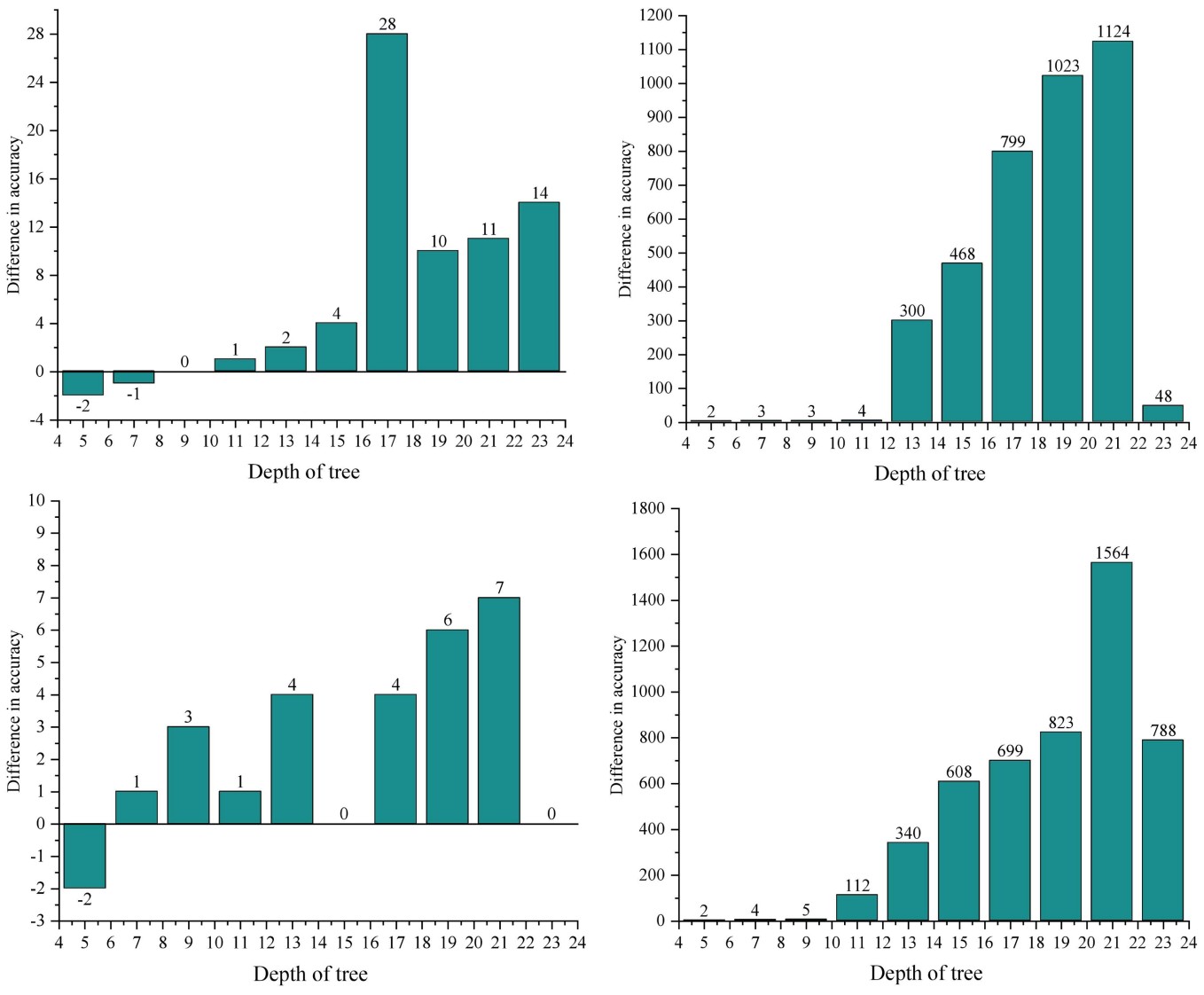

**Fig 4. Prediction performance compared with the CART algorithm.** (1) Under the UN Comtradel data set. (2) Under the OECD data set.

This suggests that, in some scenarios, the optimal decision tree algorithm may be preferable due to its simpler and more interpretable model.

Although the running time of the random forest algorithm is not as fast as that of the CART algorithm, it remains within an acceptable range, with most datasets processed in under one minute. It was also observed during the experiments that variations in hyperparameter tuning can lead to different running times for the random forest algorithm.

### 4.5 Ablation experiments

To analyze the model's performance in resource allocation for trade activity checking, this section employs ablation experiments for comparative analysis. Specifically, it involves removing dataset consistency processing and the dp-OBDT algorithm to evaluate three different resource allocation methods. For ease of reference, the final model presented in this

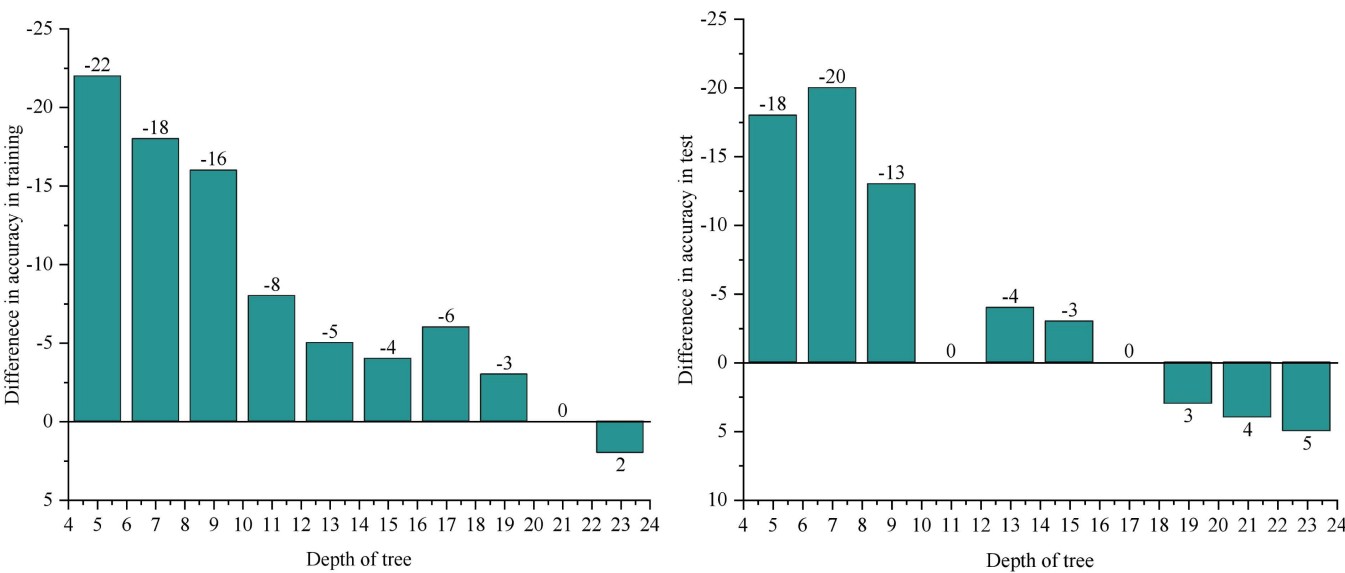

**Fig 5. Prediction performance compared with random forest algorithm.**

paper is denoted as T1, the model with dataset consistency processing removed is denoted as T2, and the model with the dp-OBDT algorithm removed is denoted as T3. The comparison of these three methods focuses on data stacking and resource utilization, as shown in Fig 6.

The prediction model estimates future data volumes based on historical data. For simplicity, predictions are based on data from time points t-2, t-1, and t. Resource allocation is evaluated against a static scheduling baseline, with subsequent calculations for resource expansion or contraction proportionate to the predicted data volume.

SubFigure (a) in Fig 6 illustrates the data pileup between the producer and the consumer, with the horizontal axis representing time and the vertical axis representing data pileup size (i.e., the number of tuples). It is evident that T3's resource allocation method exhibits fluctuating data pileup throughout the runtime, with smaller pileups when data injection is low

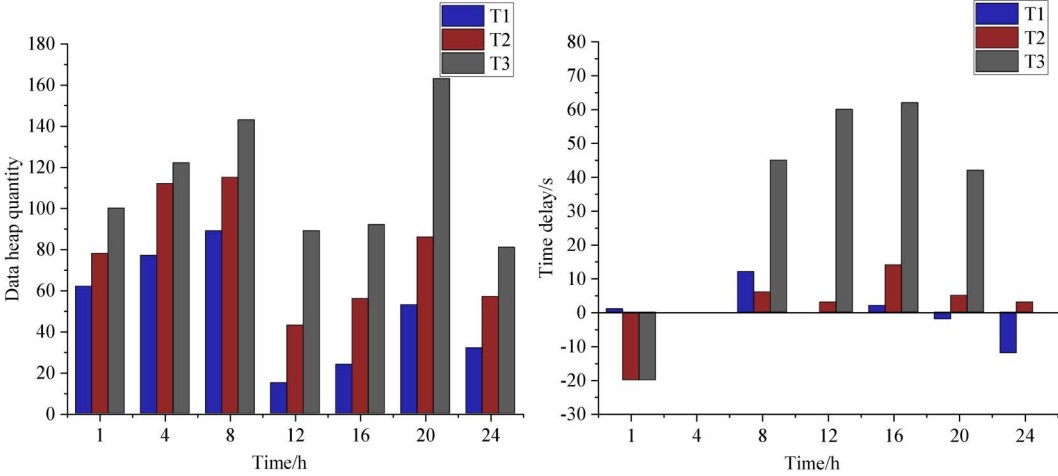

**Fig 6. Ablation experiments of data accumulation and resource utilization.**

and larger pileups when data injection is high. In contrast, T1's dynamic resource allocation method maintains lower data pileup levels with smaller fluctuations compared to static resource allocation. This is attributed to the system's ability to dynamically adjust processing capacity according to data volume, resulting in less variability.

In T2, the data pileup fluctuation is comparable to T1's dynamic scheduling, with better performance at certain times and poorer performance at others, reflecting the accuracy of predictions. For instance, before 20 hours, T2 shows significant fluctuations in data pileup, but after 20 hours, fluctuations decrease, resulting in lower pileup than in the dynamic resource allocation method. This improvement is due to the system's learning phase, where the algorithm continuously refines its parameters. Once optimized, the system allocates computational resources more effectively based on data injection rates, reducing data pileup.

SubFigure (b) evaluates delay time, with the horizontal axis representing time and the vertical axis representing delay size. Positive values indicate delays greater than one time interval, while negative values indicate processing times within one interval. Throughout the running period, T3's resource allocation method demonstrates instability, with high and low delays due to the lack of resource adjustment. After 20 hours, T3's delay becomes more stable and lower than the dynamic resource allocation method. This stability is attributed to the reactive nature of T1's dynamic adjustment. The average delay for T2 is 12.26, while T1's dynamic resource allocation method has an average delay of 15.46, which is 14.67% less than T2's delay.

Fig 7 illustrates the comparison of resource utilization and throughput. The Figure demonstrates that once the decision tree reaches a certain stage, both throughput and resource utilization improve. Experimental calculations show that T2 achieves an average resource utilization of 73.63% and a throughput of 382.71. In comparison, T1 has an average resource utilization of 60.29% and a throughput of 336.5. Relative to T3, T2 exhibits a 22% increase in average resource utilization and a 13.7% increase in throughput. This improvement is attributed to the data consistency processing and intelligent decision tree, which enable the efficient allocation of computing resources to manage fluctuating data effectively, leading to better resource utilization and throughput.

To further validate the reliability of the performance improvements achieved by the proposed model, we conducted independent two-sample t-tests on two key evaluation metrics—resource utilization and system throughput—between the full-featured model (T1) and the variant without the optimal decision tree (T3). Each configuration was executed 1,000 times to ensure statistical robustness, and the mean and standard deviation were recorded. For resource utilization, the T1 model achieved a mean of 60.29% ($\sigma=2.1$), while T3 yielded a significantly lower mean of 49.38% ($\sigma=2.4$). The resulting t-statistic was 78.42, with a p-value $<1\times10^{-26}$, indicating an extremely significant difference. Similarly, for system

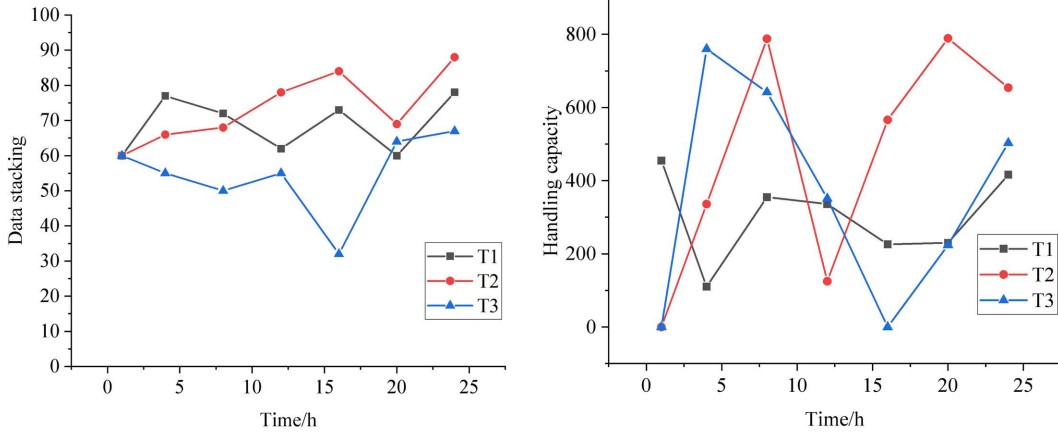

**Fig 7. Ablation experiments of resource utilization and throughput.**

throughput, T1 recorded an average of 336.5 Mbps ($\sigma = 10.2$), whereas T3 attained 295.9 Mbps ($\sigma = 11.1$). The t-statistic was 65.03, with a p-value $< 1 \times 10^{-14}$, confirming a statistically significant performance improvement.

These results clearly demonstrate that the improvements brought by the integration of the event-driven architecture and the optimal binary decision tree are not only practically effective but also statistically significant. The low p-values strongly reject the null hypothesis, affirming that the observed gains in system performance are unlikely to be due to random variation, thereby supporting the validity and generalizability of the proposed model.

### 4.6 Discussion

A deeper analysis of the experimental results presented in Sections 4.3–4.5 demonstrates that the proposed model exhibits significant advantages in runtime efficiency, providing a solid foundation for real-time monitoring and instant feedback in trade activities. The experiments show that the substantial reduction in execution time enables the system to promptly respond to various events in trade scenarios. When early signs of potential environmental risks—such as excessive carbon emissions—emerge, the system can quickly identify these issues and trigger appropriate response mechanisms to contain negative impacts in a timely manner. This capability is crucial for supporting the green and sustainable development of trade operations.

Moreover, the shorter runtime enhances the decision-making speed of the intelligent decision tree, offering timely and reliable guidance for adaptive adjustments and optimizations in trade activities. This accelerates the transition of trade processes toward a sustainable development path.

From the perspective of throughput, the high-throughput characteristics of the system allow for the full utilization of computing resources, avoiding idleness and waste, and thereby promoting carbon emission reductions at the resource usage level. The observed reduction in data backlog during experiments not only decreases data processing latency and improves system responsiveness but also frees up storage capacity, optimizing storage resource allocation and minimizing emissions associated with data storage. Additionally, high-quality data inputs enhance the accuracy of the intelligent decision tree, resulting in decisions that better reflect actual demand and resource conditions. This enables dynamic optimization of resource allocation, ensuring that while maintaining the normal operation of trade activities, the system can precisely reduce unnecessary resource consumption. Ultimately, this leads to lower carbon emissions and supports energy conservation and emission reduction goals, offering comprehensive technical support for the sustainable development of trade.

### 4.7 Potential implications of study

This study presents a sustainable trade activity monitoring model that integrates event-driven architecture with intelligent decision trees. The model demonstrates strong performance in both theoretical development and experimental validation, with broad potential for cross-domain applications. Its adaptive capabilities make it well-suited for intelligent support in dynamic, complex, and high-concurrency environments.

In intelligent manufacturing and the Industrial IoT [25,26], where heterogeneous devices, real-time data streams, and multi-source events are prevalent, the model enhances system responsiveness and decision-making by enabling event-based sensing and adaptive scheduling. This supports proactive optimization in equipment allocation, energy efficiency, and production rhythm control.

The proposed monitoring model demonstrates substantial potential for effective deployment in real-world domains. For instance, the dynamic event-driven allocation strategy aligns closely with challenges in airport slot management, where tree-structured capacity planning and adaptive allocation mechanisms can substantially reduce congestion and enhance operational efficiency [27]. The parallel between air traffic flow and trade activity streams highlights the universal importance of synchronizing resource injection and processing rates. Just as the proposed model achieves consistency between upstream and downstream applications, airport slot management requires a balance between slot allocation

and real-time passenger or cargo flow. The ability of the model to dynamically adjust computational resources mirrors the adaptive allocation of limited airport capacity, thereby ensuring system resilience under fluctuating demand conditions.

Similarly, the decentralized decision-making capability of the model resonates with autonomous satellite management, particularly in the context of pivoting path planning and reconfigurable satellite coordination [28]. In this domain, satellites must autonomously reconfigure their trajectories to adapt to unforeseen environmental changes, resource limitations, or mission requirements. The event-driven and decision-tree-based logic embedded in the proposed monitoring framework offers a blueprint for such adaptive responses. Specifically, the model's intelligent agent–driven allocation mechanism can be extended to satellite clusters, enabling real-time coordination and distributed optimization in highly dynamic environments. This analogy underscores the generalizability of the model across systems characterized by resource constraints, event-driven triggers, and high concurrency requirements.

## 5. Conclusion

This paper proposes a novel monitoring model for sustainable trade activities by integrating an event-driven architecture with an intelligent decision tree. Methodologically, the model ensures data consistency, reserves computing resources through dynamic programming for optimal binary decision tree construction, and employs intelligent agents for dynamic decision-making, enhancing adaptability to complex trade environments. The intelligent decision tree builds a fine-grained model that comprehensively considers multiple factors to provide scientifically grounded resource allocation strategies. Under the event-driven architecture, the model can respond rapidly to events, flexibly adjust resource distribution, prioritize critical trade activities, and improve overall efficiency and sustainability.

Experimental results confirm the effectiveness of the proposed model. It demonstrates high monitoring accuracy, capable of capturing critical information with precision. Its decision-making process is transparent and traceable, with strong credibility and trustworthiness that foster user confidence. The model also exhibits strong stability, ensuring reliable and continuous system operation. Overall, it offers an efficient, intelligent, and dependable solution for monitoring sustainable trade activities.

Future work will focus on algorithmic optimization to improve the adaptability and accuracy of the intelligent decision tree in more complex trade scenarios. Additionally, efforts will be made to incorporate more real-time data sources and event types, enhancing the model's situational awareness and responsiveness. By integrating machine learning and artificial intelligence technologies, the model aims to achieve smarter resource allocation and decision support, contributing to the long-term development of sustainable trade activities..

## Acknowledgments

This work is funded by 2024 Hunan Natural Science Foundation Project "Data Elements Embedded in Entity Value Chain: Empowerment and Risk", the project number is 2024JJ6199; and supported by" Scientific Research Fund of Hunan Provincial Education Department: Research on the Development of Deep Integration of Intelligent, Service-oriented and Green Manufacturing Industries in Hunan Province in the Context of Digitization", the project number is 23C0408.

## Author contributions

**Conceptualization:** Yixian Wen.

**Data curation:** Suo Zhang.

**Funding acquisition:** Yixian Wen, Suo Zhang, Sisi Zhang.

**Resources:** Sisi Zhang.

**Supervision:** Sisi Zhang.

**Writing – original draft:** Yixian Wen.

**Writing – review & editing:** Suo Zhang.

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
