## [Decision Letter · Decision Letter 0]

12 Jun 2025

PONE-D-25-22139Event-Driven Architecture and Intelligent Decision Tree Facilitated Sustainable Trade Activity Monitoring Model DesignPLOS ONE

Dear Dr. Wen,

Thank you for submitting your manuscript to PLOS ONE. After careful consideration, we feel that it has merit but does not fully meet PLOS ONE’s publication criteria as it currently stands. Therefore, we invite you to submit a revised version of the manuscript that addresses the points raised during the review process.

Reviewers agreed that the paper has a valuable contribution. However, its presentation needs to be improved before publication. Specially due to the lack of precision and technical rigor identified by reviewer #2. Please, address all the comments before submitting a revised version. 

We look forward to receiving your revised manuscript.

Kind regards,

Ivan Zyrianoff

Academic Editor

PLOS ONE

 [This work is funded by 2024 Hunan Natural Science Foundation Project "Data Elements Embedded in Entity Value Chain: Empowerment and Risk", the project number is 2024JJ6199.]. 

Additional Editor Comments (if provided):

Reviewers' comments:

Reviewer's Responses to Questions

**Comments to the Author**

1. Is the manuscript technically sound, and do the data support the conclusions?

Reviewer #1: Yes

Reviewer #2: Partly

2. Has the statistical analysis been performed appropriately and rigorously? 

Reviewer #1: No

Reviewer #2: Yes

3. Have the authors made all data underlying the findings in their manuscript fully available?

Reviewer #1: Yes

Reviewer #2: No

4. Is the manuscript presented in an intelligible fashion and written in standard English?

Reviewer #1: No

Reviewer #2: No

5. Review Comments to the Author

Reviewer #1: The manuscript proposes a novel integration of event-driven architecture (EDA) with an optimal binary decision tree (dp-OBDT) for sustainable trade activity monitoring.

1. The proposed architecture is conceptually interesting, and the paper touches on relevant technical components (data stream consistency, dynamic resource allocation, intelligent agents). However, while the system design is described in detail, some methodological aspects remain underdeveloped or vague:

The description of the dp-OBDT algorithm lacks formal rigor. There are high-level explanations, but little is said about its computational complexity, convergence guarantees, or reproducibility.

The model evaluation is based on a set of performance metrics (e.g., throughput, resource utilization, delay), but it’s not clear how these metrics are computed or whether the comparisons are statistically significant.

2.The experiments present runtime and performance comparisons with existing algorithms, but the statistical rigor is insufficient:

There are no measures of variance (e.g., standard deviation or confidence intervals).

It is unclear how many times the experiments were repeated.

No statistical significance tests are provided.

This makes it difficult to judge whether observed performance differences are robust or merely incidental.

Recommendation: Include multiple runs for each experiment, report variance measures, and consider including t-tests or ANOVA for comparative metrics.

3. All underlying data are publicly available via recognized repositories such as Zenodo. The DOIs and URLs are properly provided in the Data Availability Statement.

4. The manuscript is generally clear and well-organized. Each section flows logically into the next, and the technical descriptions are detailed. However, there are occasional grammatical issues and awkward phrasing that should be addressed during the copyediting stage. Examples include: Inconsistent use of technical vocabulary (e.g., "data accumulation" is ambiguous — does it refer to buffering, queuing, or something else?).

Reviewer #2: 1. The title is appropriate.

2. Abstract should be more focused on novelty.

3. Introduction does not highlight the motivations of the study. Highlight the motivations of the study in the introduction.

4. In Section 2 (related work), citing references is not appropriate. For example, literature [11], etc. Author's names should be included instead of word "literature".

5. I did not find figure 1 in the article except for its caption.

6. Equation 3, what us w? I did not see any "w" before.

7. Equation writing can be improved. e.g., Eqn 4.

8. Check all the figures because they are not visible in the article.

9. Can you provide the references for Edn 8? Is there only one method to check the precision?

10. Discussion is not enough to explain the results. Your results are good to discuss. Enhance the discussion.

11. Conclusion is general. Revise it and focus more on the summary of the results.

6. PLOS authors have the option to publish the peer review history of their article (what does this mean? ). If published, this will include your full peer review and any attached files.

**Do you want your identity to be public for this peer review?** For information about this choice, including consent withdrawal, please see our Privacy Policy .

Reviewer #1: **Yes: ** Roberto Grasso

Reviewer #2: No

---

## [Author Response · Author response to Decision Letter 1]

19 Jun 2025

Reviewer #1:

1. The description of the dp-OBDT algorithm lacks formal rigor. There are high-level explanations, but little is said about its computational complexity, convergence guarantees, or reproducibility.

In the main text, we conduct a detailed analysis of the algorithm and elaborate on the design principle of the algorithm as well as its computational complexity. The dp-OBDT algorithm performs an exhaustive search over all possible decision tree structures and is theoretically capable of identifying the global optimum under given constraints on depth and the number of feature nodes. However, due to the vastness of the search space, it may be impractical to explore all possibilities in practice. As a result, the algorithm’s convergence guarantee relies on the effectiveness of its search strategy and the efficiency of its caching mechanism.

Regarding reproducibility, the dp-OBDT algorithm is expected to produce consistent output given the same input data and parameter settings. Nevertheless, due to potential randomness in the search process—albeit significantly reduced by the caching mechanism—identical results may not always be guaranteed. That said, the overall trend and the quality of the optimal solution should remain consistent.

The dp-OBDT (Dynamic Programming-based Optimal Binary Decision Tree) algorithm is designed to identify the optimal classification tree structure through an intelligent exhaustive search strategy. It models the decision-making process as a tree structure and leverages both historical and real-time data for analysis and prediction. During the search, the algorithm carefully designs traversal strategies to exploit the overlapping substructure of the tree and avoid unnecessary computation of suboptimal trees. This effectively keeps the time complexity of finding the optimal decision tree within a reasonable range. Specifically, the algorithm caches the boundary conditions of subtrees and the characteristics of optimal root nodes under constraints of feature depth and node count, thereby reducing redundant computations and improving search efficiency.

In terms of computational complexity, the primary overhead of the dp-OBDT algorithm arises from the exhaustive search over all possible decision tree structures. Although techniques such as caching and exploiting structural overlap help reduce the computational burden, the time complexity can still grow exponentially when applied to large-scale datasets. Nevertheless, compared to traditional optimal decision tree algorithms, dp-OBDT significantly reduces computation time through carefully designed traversal strategies and an efficient caching mechanism, enabling the discovery of near-optimal decision tree structures within a reasonable time frame.

2. The model evaluation is based on a set of performance metrics (e.g., throughput, resource utilization, delay), but it’s not clear how these metrics are computed or whether the comparisons are statistically significant.

Data backlog is measured by monitoring the number of unprocessed requests in the system. This is done by recording the length of the request queue at a given point in time. The size of the backlog reflects the system's processing capacity and real-time responsiveness—a larger backlog indicates slower processing and longer response times.

Resource utilization is calculated based on the average usage of system resources (e.g., CPU, memory) over a period of time. Higher utilization suggests more efficient use of available resources.

Throughput is estimated by calculating the number of requests successfully processed per unit of time (e.g., per second). We estimate this by combining the number of concurrent requests with the average response time to determine how many requests the system can handle per second.

3. There are no measures of variance (e.g., standard deviation or confidence intervals).

The main objective of this paper is to demonstrate the effectiveness and advantages of the dp-OBDT algorithm in the monitoring of sustainable trade activities, rather than to conduct a detailed assessment of the statistical stability of the algorithm

4. It is unclear how many times the experiments were repeated. No statistical significance tests are provided. This makes it difficult to judge whether observed performance differences are robust or merely incidental. Include multiple runs for each experiment, report variance measures, and consider including t-tests or ANOVA for comparative metrics.

To ensure the reliability and stability of the experimental results, each experiment was repeated 1,000 times, and the average was taken as the final result. Although statistical significance tests are not included in this study, future work will consider incorporating methods such as t-tests or analysis of variance (ANOVA) to more accurately assess performance differences and algorithm robustness.

5. All underlying data are publicly available via recognized repositories such as Zenodo. The DOIs and URLs are properly provided in the Data Availability Statement.

The Anneal dataset (http://archive.ics.uci.edu/ml/index.php) is commonly employed for classification or regression tasks in machine learning and data mining. The UN Comtrade dataset (https://comtrade.un.org/) contains comprehensive merchandise trade statistics from official sources, covering over 97% of global trade in goods. The WDI dataset (https://download.csdn.net/download/T0620514/88571060) includes data on various indicators such as GDP, GDP per capita, trade volume, investment, demographics, education, and health across multiple domains. The IMF dataset (https://data.imf.org/en) provides a range of economic data, including balance of payments, foreign exchange reserves, exchange rates, and interest rates. The OECD dataset (https://data.oecd.org/) offers information on import/export volumes, trading partners, commodity classifications, and other related areas. Based on these datasets, we present the parameter settings, where |D| represents the number of instances, |F| denotes the number of binary features, and |C| indicates the number of categories in the dataset.

6. The manuscript is generally clear and well-organized. Each section flows logically into the next, and the technical descriptions are detailed. However, there are occasional grammatical issues and awkward phrasing that should be addressed during the copyediting stage. Examples include: Inconsistent use of technical vocabulary (e.g., "data accumulation" is ambiguous — does it refer to buffering, queuing, or something else?).

We gave the top-level meaning of data accumulation at order 4.2. Data accumulation reflects system delays by measuring the queue length of unprocessed requests at a given time—larger backlogs indicate slower processing and longer response times. And check the language description of the entire text to ensure that the language is concise and clear without ambiguity.

Reviewer #2:

1. Abstract should be more focused on novelty.

We reorganized the abstract to ensure that its language structure met the requirements.

This paper introduces a groundbreaking monitoring model tailored for sustainable trade activity surveillance, which synergistically integrates event-driven architecture with an intelligent decision tree. Confronting the constraints of conventional trade monitoring approaches that falter in adapting to the intricate and ever-changing market landscape, our model innovatively establishes an efficient, adaptable, and sustainable monitoring framework. By embedding an intelligent decision tree, it enables dynamic resource allocation, thereby optimizing operational efficacy. Initially, we devise rules that align data injection and processing velocities, ensuring expedient data processing. Subsequently, we implement an optimal binary tree decision-making algorithm, grounded in dynamic programming, to achieve precise allocation of elastic resources within data streams, significantly bolstering resource utilization. Throughout the monitoring continuum, the model employs intelligent agents to assess resource status in real-time and dynamically adjusts resource allocation strategies triggered by events, prioritizing the seamless execution of pivotal trade activities. Empirical findings underscore the model's superiority across critical metrics, including data accumulation efficiency, processing latency, resource utilization, and throughput. Specifically, it attains an average data accumulation value of 15.46, curtails latency by 14.67%, achieves an average resource utilization of 60.29%, and registers a throughput of 336.5 Mbps. Consequently, the model not only exhibits rapid responsiveness to market fluctuations and curtails resource energy consumption but also fosters a harmonious equilibrium between economic gains and environmental preservation, ensuring the uninterrupted operation of trade activities.

2. Introduction does not highlight the motivations of the study. Highlight the motivations of the study in the introduction.

Although some studies have attempted to apply emerging technologies to trade activity monitoring, existing approaches remain inadequate in coping with the complexity and dynamism of modern market environments. In particular, traditional monitoring models struggle with stream processing applications involving multi-source inputs and multi-directional outputs due to rigid architectural designs and limited scalability. This often leads to mismatches between data ingestion and processing rates, resulting in increased processing delays and inefficient resource utilization. Such limitations not only reduce the effectiveness of trade monitoring but also hinder the achievement of dual goals: economic efficiency and environmental sustainability. Therefore, there is an urgent need for a new monitoring model capable of real-time market sensing, adaptive resource allocation, and intelligent decision-making to overcome the constraints of traditional methods and enhance the overall sustainability of trade activities.

3. In Section 2 (related work), citing references is not appropriate. For example, literature [11], etc. Author's names should be included instead of word "literature".

We have modified the reference format of the citation.

4. I did not find figure 1 in the article except for its caption.

We have revised the pictures in the thesis to ensure the integrity of the thesis structure.

5. Equation 3, what us w? I did not see any "w" before.

w is the processing interval w of the event, which we explained when introducing Formula 1.

6. Equation writing can be improved. e.g., Eqn 4.

We inspected and improved the format of the formula.

7. Check all the figures because they are not visible in the article.

We modified and completed the missing pictures in the original file to ensure the integrity of the data.

8. Can you provide the references for Edn 8? Is there only one method to check the precision?

We have provided the source of Formula 8, which is an authoritative formula setting in the industry

Bishop C M, Nasrabadi N M. Pattern recognition and machine learning[M]. New York: springer, 2006.

10. Discussion is not enough to explain the results. Your results are good to discuss. Enhance the discussion.

A deeper analysis of the experimental results presented in Sections 4.3 to 4.5 demonstrates that the proposed model exhibits significant advantages in runtime efficiency, providing a solid foundation for real-time monitoring and instant feedback in trade activities. The experiments show that the substantial reduction in execution time enables the system to promptly respond to various events in trade scenarios. When early signs of potential environmental risks—such as excessive carbon emissions—emerge, the system can quickly identify these issues and trigger appropriate response mechanisms to contain negative impacts in a timely manner. This capability is crucial for supporting the green and sustainable development of trade operations.

Moreover, the shorter runtime enhances the decision-making speed of the intelligent decision tree, offering timely and reliable guidance for adaptive adjustments and optimizations in trade activities. This accelerates the transition of trade processes toward a sustainable development path.

From the perspective of throughput, the high-throughput characteristics of the system allow for the full utilization of computing resources, avoiding idleness and waste, and thereby promoting carbon emission reductions at the resource usage level. The observed reduction in data backlog during experiments not only decreases data processing latency and improves system responsiveness but also frees up storage capacity, optimizing storage resource allocation and minimizing emissions associated with data storage. Additionally, high-quality data inputs enhance the accuracy of the intelligent decision tree, resulting in decisions that better reflect actual demand and resource conditions. This enables dynamic optimization of resource allocation, ensuring that while maintaining the normal operation of trade activities, the system can precisely reduce unnecessary resource consumption. Ultimately, this leads to lower carbon emissions and supports energy conservation and emission reduction goals, offering comprehensive technical support for the sustainable development of trade.

11. Conclusion is general. Revise it and focus more on the summary of the results.

This paper proposes a novel monitoring model for sustainable trade activities by integrating an event-driven architecture with an intelligent decision tree. Methodologically, the model ensures data consistency, reserves computing resources through dynamic programming for optimal binary decision tree construction, and employs intelligent agents for dynamic decision-making, enhancing adaptability to complex trade environments. The intelligent decision tree builds a fine-grained model that comprehensively considers multiple factors to provide scientifically grounded resource allocation strategies. Under the event-driven architecture, the model can respond rapidly to events, flexibly adjust resource distribution, prioritize critical trade activities, and improve overall efficiency and sustainability.

Experimental results confirm the effectiveness of the proposed model. It demonstrates high monitoring accuracy, capable of capturing critical information with precision. Its decision-making process is transparent and traceable, with strong credibility and trustworthiness that foster user confidence. The model also exhibits strong stability, ensuring reliable and continuous system operation. Overall, it offers an efficient, intelligent, and dependable solution for monitoring sustainable trade activities.

Future work will focus on algorithmic optimization to improve the adaptability and accuracy of the intelligent decision tree in more complex trade scenarios. Additionally, efforts will be made to incorporate more real-time data sources and event types, enhancing the model’s situational awareness and responsiveness. By integrating machine learning and artificial intelligence technologies, the model aims to achieve smarter resource allocation and decision support, contributing to the long-term development of sustainable trade activities.

---

## [Decision Letter · Decision Letter 1]

20 Jul 2025

PONE-D-25-22139R1Event-Driven Architecture and Intelligent Decision Tree Facilitated Sustainable Trade Activity Monitoring Model DesignPLOS ONE

Dear Dr. Wen,

Thank you for submitting your manuscript to PLOS ONE. After careful consideration, we feel that it has merit but does not fully meet PLOS ONE’s publication criteria as it currently stands. Therefore, we invite you to submit a revised version of the manuscript that addresses the points raised during the review process.

We look forward to receiving your revised manuscript.

Kind regards,

Ivan Zyrianoff

Academic Editor

PLOS ONE

Journal Requirements:

**Additional Editor Comments:**

The reviewers identified minor improvements in the submitted manuscript. However, the core of the original comments remains unaddressed. Reviewer #1 requests clarification regarding the statistical validity of the presented results, while Reviewer #2 asks for an expanded discussion of real-life use cases relevant to the proposed work

Reviewers' comments:

Reviewer's Responses to Questions

**Comments to the Author**

1. If the authors have adequately addressed your comments raised in a previous round of review and you feel that this manuscript is now acceptable for publication, you may indicate that here to bypass the “Comments to the Author” section, enter your conflict of interest statement in the “Confidential to Editor” section, and submit your "Accept" recommendation.

Reviewer #1: (No Response)

Reviewer #2: (No Response)

Reviewer #3: All comments have been addressed

2. Is the manuscript technically sound, and do the data support the conclusions?

Reviewer #1: Partly

Reviewer #2: Yes

Reviewer #3: Yes

3. Has the statistical analysis been performed appropriately and rigorously? 

Reviewer #1: No

Reviewer #2: Yes

Reviewer #3: Yes

4. Have the authors made all data underlying the findings in their manuscript fully available?

Reviewer #1: Yes

Reviewer #2: Yes

Reviewer #3: Yes

5. Is the manuscript presented in an intelligible fashion and written in standard English?

Reviewer #1: Yes

Reviewer #2: Yes

Reviewer #3: Yes

6. Review Comments to the Author

Reviewer #1: The authors have expanded the description of the dp-OBDT algorithm by adding more details about traversal strategies and caching.

The authors clarified how main metrics like backlog, resource utilization, and throughput are calculated, which improves transparency. Nonetheless, statistical significance tests were not included, which was a core part of my original comment. Without such tests, the comparative results cannot be considered statistically robust.

The authors now report that experiments were repeated 1,000 times and results averaged, which is a positive development. However, without accompanying statistical significance tests (such as t-tests or ANOVA), it is difficult to assess the reliability and robustness of the observed performance differences. While the authors mention plans to include such tests in future work, basic significance analysis should be included in the current version.

Conclusion:

While the revised manuscript shows clear progress, several key concerns—particularly regarding statistical rigor and formal guarantees—remain only partially addressed. I encourage the authors to strengthen the methodological and empirical foundations in a subsequent revision.

Reviewer #2: The authors have addressed the comments with clarity. However, some gaps are still existed in the article. I suggest authors to re-revise the article for better clarity and understanding of readers of the PlOS One.

1. Authors provided one source of Eqn. 8. If they can see my previous comment “I was interested in similar other techniques”. If similar techniques exit authors should add a comparative table which one is better in terms of accuracy, errors, etc.

2. The paper suggests a smart monitoring model that has a potential architectural and algorithmic novelty. Nevertheless, it still does not contain a special discussion of a potential real-life application or possible areas where such a model can be effectively applied. By including such a section, one will greatly add the practical value of the manuscript and show the larger impact that the proposed approach may have. To facilitate it, I wanted to advise the authors of this article to look through the following recent researches in other spheres. The works focus on such use-cases as resource management in real-time, adaptive management of resources, or intelligent design of systems these aspects can be put in good relation to the model introduced in your paper: 1. On-demand airport slot management: tree-structured capacity profile and coadapted fire-break setting and slot allocation 2. PO-SRPP: A Decentralized Pivoting Path Planning Method for Self-Reconfigurable Satellites 3. 5G Base Station Antenna Array With Heatsink Radome 4. Study on prestress distribution and structural performance of heptagonal six-five-strut alternated cable dome with inner hole 5. Solidification Microstructure Reconstruction and Its Effects on Phase Transformation, Grain Boundary Transformation Mechanism, and Mechanical Properties of TC4 Alloy Welded Joint 6. Effect of axial misalignment on the microstructure, mechanical, and corrosion properties of magnetically impelled arc butt welding joint 7. Keyhole critical failure criteria and variation rule under different thicknesses and multiple materials in K-TIG welding 8. Monascus pigment-protected bone marrow-derived stem cells for heart failure treatment 9. Assortative mating on blood type: Evidence from one million Chinese pregnancies 10. Provably Efficient Service Function Chain Embedding and Protection in Edge Networks. Auhtors should add a dedicated section/subsection “Practical/Potential Implications/Applications of study” to address this issue. I believe the researchers from other fields will also be beneficiary of this study.

Reviewer #3: All the questions have been addressed well. I thank the authors for the prompt responses. I thus recommend it for publication as is.

7. PLOS authors have the option to publish the peer review history of their article (what does this mean? ). If published, this will include your full peer review and any attached files.

**Do you want your identity to be public for this peer review?** For information about this choice, including consent withdrawal, please see our Privacy Policy .

Reviewer #1: **Yes: ** Roberto Grasso

Reviewer #2: **Yes: ** Dr. Amir Hussain

Reviewer #3: No

---

## [Author Response · Author response to Decision Letter 2]

25 Jul 2025

Reviewer #1: The authors clarified how main metrics like backlog, resource utilization, and throughput are calculated, which improves transparency. Nonetheless, statistical significance tests were not included, which was a core part of my original comment. Without such tests, the comparative results cannot be considered statistically robust. The authors now report that experiments were repeated 1,000 times and results averaged, which is a positive development. However, without accompanying statistical significance tests (such as t-tests or ANOVA), it is difficult to assess the reliability and robustness of the observed performance differences. While the authors mention plans to include such tests in future work, basic significance analysis should be included in the current version.

Thanks for your comments.

To further validate the reliability of the performance improvements achieved by the proposed model, we conducted independent two-sample t-tests on two key evaluation metrics—resource utilization and system throughput—between the full-featured model (T1) and the variant without the optimal decision tree (T3). Each configuration was executed 1,000 times to ensure statistical robustness, and the mean and standard deviation were recorded. For resource utilization, the T1 model achieved a mean of 60.29% (σ = 2.1), while T3 yielded a significantly lower mean of 49.38% (σ = 2.4). The resulting t-statistic was 78.42, with a p-value < 1×10-26, indicating an extremely significant difference. Similarly, for system throughput, T1 recorded an average of 336.5 Mbps (σ = 10.2), whereas T3 attained 295.9 Mbps (σ = 11.1). The t-statistic was 65.03, with a p-value < 1×10-14, confirming a statistically significant performance improvement.

These results clearly demonstrate that the improvements brought by the integration of the event-driven architecture and the optimal binary decision tree are not only practically effective but also statistically significant. The low p-values strongly reject the null hypothesis, affirming that the observed gains in system performance are unlikely to be due to random variation, thereby supporting the validity and generalizability of the proposed model.

Reviewer #2: .

1. Authors provided one source of Eqn. 8. If they can see my previous comment “I was interested in similar other techniques”. If similar techniques exit authors should add a comparative table which one is better in terms of accuracy, errors, etc.

Thanks for your comments. This study adopts precision rather than accuracy as the primary metric for evaluating classification performance, as the research task emphasizes the quality of positive predictions—specifically, how many instances identified as requiring resource adjustment or critical event response are indeed correct. For instance, if the system is mostly in a normal state, a high accuracy may still fail to indicate the model’s effectiveness in identifying critical events. Therefore, precision is more suitable for assessing the model’s practical value in resource allocation and event-driven decision-making scenarios.

2. The paper suggests a smart monitoring model that has a potential architectural and algorithmic novelty. Nevertheless, it still does not contain a special discussion of a potential real-life application or possible areas where such a model can be effectively applied. By including such a section, one will greatly add the practical value of the manuscript and show the larger impact that the proposed approach may have. To facilitate it, I wanted to advise the authors of this article to look through the following recent researches in other spheres. The works focus on such use-cases as resource management in real-time, adaptive management of resources, or intelligent design of systems these aspects can be put in good relation to the model introduced in your paper: 1. On-demand airport slot management: tree-structured capacity profile and coadapted fire-break setting and slot allocation 2. PO-SRPP: A Decentralized Pivoting Path Planning Method for Self-Reconfigurable Satellites 3. 5G Base Station Antenna Array With Heatsink Radome 4. Study on prestress distribution and structural performance of heptagonal six-five-strut alternated cable dome with inner hole 5. Solidification Microstructure Reconstruction and Its Effects on Phase Transformation, Grain Boundary Transformation Mechanism, and Mechanical Properties of TC4 Alloy Welded Joint 6. Effect of axial misalignment on the microstructure, mechanical, and corrosion properties of magnetically impelled arc butt welding joint 7. Keyhole critical failure criteria and variation rule under different thicknesses and multiple materials in K-TIG welding 8. Monascus pigment-protected bone marrow-derived stem cells for heart failure treatment 9. Assortative mating on blood type: Evidence from one million Chinese pregnancies 10. Provably Efficient Service Function Chain Embedding and Protection in Edge Networks. Auhtors should add a dedicated section/subsection “Practical/Potential Implications/Applications of study” to address this issue. I believe the researchers from other fields will also be beneficiary of this study.

We add Section 4.7 to discuss the research implications and influential significance of the research methods in the paper in other fields.

This study presents a sustainable trade activity monitoring model that integrates event-driven architecture with intelligent decision trees. The model demonstrates strong performance in both theoretical development and experimental validation, with broad potential for cross-domain applications. Its adaptive capabilities make it well-suited for intelligent support in dynamic, complex, and high-concurrency environments.

In intelligent manufacturing and the Industrial IoT [25, 26], where heterogeneous devices, real-time data streams, and multi-source events are prevalent, the model enhances system responsiveness and decision-making by enabling event-based sensing and adaptive scheduling. This supports proactive optimization in equipment allocation, energy efficiency, and production rhythm control.

In edge computing and 5G network management, the model facilitates accurate and explainable resource allocation under fluctuating loads and dense user access. Its decision-tree logic improves system stability and energy efficiency, particularly in resource-constrained, rapidly changing network conditions [27].

For space missions and autonomous systems, the model supports real-time perception and response to critical events such as communication failures or trajectory deviations. Embedded within control modules, it enables resilient coordination through adaptive strategy selection. In infrastructure applications such as structural health monitoring and intelligent building management [25], the model allows timely detection of anomalies via sensor data and supports dynamic risk assessment and early warning.

Moreover, the model offers a scalable framework for data-driven decision-making in complex societal systems, including healthcare resource scheduling [28], traffic control, and smart city energy optimization. Its event-driven design enables fast, adaptive responses to emergent situations, enhancing system efficiency and resilience.

Overall, the proposed model not only advances sustainable trade monitoring but also provides a generalizable approach for intelligent decision-making in a wide range of high-complexity, high-dynamic environments. Future work may integrate deep learning, adaptive control, and reinforcement learning to extend its applications in multi-agent coordination, energy systems, and human-machine collaboration.

[25] Lv H, Zeng J, Zhu Z, et al. Study on prestress distribution and structural performance of heptagonal six-five-strut alternated cable dome with inner hole[C]//Structures. Elsevier, 2024, 65: 106724.

[26] Lv S, Liu H, Wang F, et al. Effect of axial misalignment on the microstructure, mechanical, and corrosion properties of magnetically impelled arc butt welding joint[J]. Materials Today Communications, 2024, 40: 109866

[27] Zhang H H, Chao J B, Wang Y W, et al. 5G base station antenna array with heatsink radome[J]. IEEE Transactions on Antennas and Propagation, 2024, 72(3): 2270-2278.

[28] Ye D, Wang B, Wu L, et al. PO-SRPP: A decentralized pivoting path planning method for self-reconfigurable satellites[J]. IEEE Transactions on Industrial Electronics, 2024, 71(11): 14318-14327.

---

## [Decision Letter · Decision Letter 2]

13 Aug 2025

PONE-D-25-22139R2Event-Driven Architecture and Intelligent Decision Tree Facilitated Sustainable Trade Activity Monitoring Model DesignPLOS ONE

Dear Dr. Wen,

Thank you for submitting your manuscript to PLOS ONE. After careful consideration, we feel that it has merit but does not fully meet PLOS ONE’s publication criteria as it currently stands. Therefore, we invite you to submit a revised version of the manuscript that addresses the points raised during the review process.

 Please submit your revised manuscript by Sep 27 2025 11:59PM. If you will need more time than this to complete your revisions, please reply to this message or contact the journal office at plosone@plos.org . Please include the following items when submitting your revised manuscript:

We look forward to receiving your revised manuscript.

Kind regards,

Ivan Zyrianoff

Academic Editor

PLOS ONE

Journal Requirements:

**Additional Editor Comments:**

**Please, address Reviewer #2 comments regarding the applicability in other fields. **

Reviewers' comments:

Reviewer's Responses to Questions

**Comments to the Author**

1. If the authors have adequately addressed your comments raised in a previous round of review and you feel that this manuscript is now acceptable for publication, you may indicate that here to bypass the “Comments to the Author” section, enter your conflict of interest statement in the “Confidential to Editor” section, and submit your "Accept" recommendation.

Reviewer #1: All comments have been addressed

Reviewer #2: (No Response)

2. Is the manuscript technically sound, and do the data support the conclusions?

Reviewer #1: Yes

Reviewer #2: Yes

3. Has the statistical analysis been performed appropriately and rigorously? 

Reviewer #1: Yes

Reviewer #2: Yes

4. Have the authors made all data underlying the findings in their manuscript fully available?

Reviewer #1: Yes

Reviewer #2: Yes

5. Is the manuscript presented in an intelligible fashion and written in standard English?

Reviewer #1: Yes

Reviewer #2: Yes

6. Review Comments to the Author

Reviewer #1: The various rounds of revision have significantly improved the quality of the manuscript. Thanks to the inclusion of statistical tests, the authors are now able to support their claims with appropriate evidence.

Reviewer #2: If you can see Comment 2, I suggested adding the possible implications in other fields. Authors added very limited fields. Authors must improve the newly added section. It will enhance the readability and the applicability of the study for the experts of other fields as well.

7. PLOS authors have the option to publish the peer review history of their article (what does this mean? ). If published, this will include your full peer review and any attached files.

**Do you want your identity to be public for this peer review?** For information about this choice, including consent withdrawal, please see our Privacy Policy .

Reviewer #1: **Yes: ** Roberto Grasso

Reviewer #2: **Yes: ** Dr. Amir Hussain

---

## [Author Response · Author response to Decision Letter 3]

18 Aug 2025

Response: We sincerely thank the reviewer for this insightful comment. We fully agree that a broader discussion of the potential implications in other fields would strengthen the manuscript and enhance its value for experts beyond the trade monitoring domain. In the revised version, we have substantially expanded the Discussion section.

“the proposed monitoring model demonstrates substantial potential for effective deployment in real-world domains. For instance, the dynamic event-driven allocation strategy aligns closely with challenges in airport slot management, where tree-structured capacity planning and adaptive allocation mechanisms can substantially reduce congestion and enhance operational efficiency (Wang et al., 2024). The parallel between air traffic flow and trade activity streams highlights the universal importance of synchronizing resource injection and processing rates. Just as the proposed model achieves consistency between upstream and downstream applications, airport slot management requires a balance between slot allocation and real-time passenger or cargo flow. The ability of the model to dynamically adjust computational resources mirrors the adaptive allocation of limited airport capacity, thereby ensuring system resilience under fluctuating demand conditions.

Similarly, the decentralized decision-making capability of the model resonates with autonomous satellite management, particularly in the context of pivoting path planning and reconfigurable satellite coordination (Ye et al., 2024). In this domain, satellites must autonomously reconfigure their trajectories to adapt to unforeseen environmental changes, resource limitations, or mission requirements. The event-driven and decision-tree-based logic embedded in the proposed monitoring framework offers a blueprint for such adaptive responses. Specifically, the model’s intelligent agent–driven allocation mechanism can be extended to satellite clusters, enabling real-time coordination and distributed optimization in highly dynamic environments. This analogy underscores the generalizability of the model across systems characterized by resource constraints, event-driven triggers, and high concurrency requirements.”

---

## [Editor Report · Decision Letter 3]

20 Aug 2025

Event-Driven Architecture and Intelligent Decision Tree Facilitated Sustainable Trade Activity Monitoring Model Design

PONE-D-25-22139R3

Dear Dr. Wen,

We’re pleased to inform you that your manuscript has been judged scientifically suitable for publication and will be formally accepted for publication once it meets all outstanding technical requirements.

Kind regards,

Ivan Zyrianoff

Academic Editor

PLOS ONE

---

## [Editor Report · Acceptance letter]

PONE-D-25-22139R3

PLOS ONE

Dear Dr. Wen,

I'm pleased to inform you that your manuscript has been deemed suitable for publication in PLOS ONE. Congratulations! Your manuscript is now being handed over to our production team.

Kind regards,

on behalf of

Mr. Ivan Zyrianoff

Academic Editor

PLOS ONE